

# Main complaints identified by parents of children with developmental delays during the initial consultation: a 10-year all-case study

Yasuaki Kusumoto[1], Eri Takahashi[1], Kenji Takaki[2], Tadamitsu Matsuda[3] and Osamu Nitta[4]

[1] Department of Physical Therapy, Fukushima Medical University School of Health Sciences, Fukushima City, Japan
[2] Major of Physical Therapy, Department of Rehabilitation, Tokyo University of Technology, Ohta-ku, Japan
[3] Department of Physical Therapy, Faculty of Health Sciences, Juntendo University, Bunkyo-ku, Japan
[4] Department of Physical Therapy, R Professional University of Rehabilitation, Tsuchiura City, Japan

Corresponding author
Yasuaki Kusumoto,
kusumoto@fmu.ac.jp

## ABSTRACT

**Background**. In Japan, the child development support initiative is one of the government's daycare support programs for children with disabilities. Children, aged 0–6 years, who are not attending elementary school can participate in the initiative and receive various support. Reports on the approaches taken by private child development support centers and the guardians' perceptions are increasing. Conversely, information from public child development support centers, which serve as places for initial developmental consultation, is extremely scarce. Moreover, there are no nationwide reports on the main complaints from each region, which are of concern to the parents. This study aimed to clarify children's gender and age, presence of referral sources, and characteristics of the main complaints obtained during the initial consultation with parents of children with developmental delays, who used a public developmental support center in a medium-sized city in Tokyo.

**Methods**. This study included 1,241 parents of children with developmental delays (average 40.3 months, range 2–87 months). Five questions regarding each child's characteristics (gender, age in months, and medical diagnosis), referral sources for the use of support centers, and main complaints that they would like to discuss at the initial consultation, were asked. The participants were asked to describe their main complaints (specific consultation details) as precisely as possible. From the free-form descriptions of the main complaints, 137 codes were extracted and grouped into 13 categories. Participants were divided into two groups according to the presence ($n = 122$) or absence ($n = 1,119$) of a medical diagnosis. The $t$-test, chi-square test, and Fisher's exact probability test were used to examine differences between the two groups. Logistic regression analysis with forced entry was performed to examine whether the factors related to the main complaints raised by parents of children with developmental delays differed depending on the presence or absence of a medical diagnosis.

**Results**. The most common chief complaint at the time of the initial consultation was "language development" (43.9%), followed by "childcare and preschool counseling" (15.4%), "hyperactivity/inattention" (13.9%), and "general developmental issues" (13.6%). The regression analysis revealed that gender, age (months), and general

developmental issues were factors associated with the presence or absence of a medical diagnosis were gender, age in months, and general developmental issues. The odds ratios (95% confidence intervals) were 1.573 (1.056–2.343) for gender, 0.988 (0.976–1.000) for age (months), and 0.421 (0.200–0.886) for general developmental issues.
**Conclusion**. Professionals involved in child development support are expected to have broad knowledge of various developmental issues as well as comprehensive knowledge of local childcare and schooling systems.

## INTRODUCTION

In Japan, according to a 2022 survey by the Ministry of Education, Culture, Sports, Science, and Technology, 8.8% of students in elementary and junior high schools receive special support. Of these, approximately 30%–40% receive special educational support and classroom guidance (*Ministry of Education, Culture, Sports, Science, and Technology, 2022*). The child development support initiative, which provides different types of support, is one of the government's daycare support programs for children with disabilities for 0- to 6-year-old children who are not yet in elementary school (*Ministry of Health, Labour and Welfare, 2017a*). The child development support initiative includes developmental consultation services, support for independence in daily life, rehabilitation, and a place to play and learn (similar to nursery school or kindergarten) to promote the child's development and reduce family anxiety (*Ministry of Health, Labour and Welfare, 2017a*). In recent years, the number of children and students receiving special support in elementary and junior high schools has increased. The demand for child development support services among preschool children, with and without a medical diagnosis, is rapidly increasing. In particular, the demand for child development support and rehabilitation services for preschool children has grown substantially.

Since the revision of the Child Welfare Law in 2012, child development support has included two types of facilities: (1) Public child development support centers, which aim to ensure quality so that children with disabilities can receive appropriate support regardless of the type of disability, and (2) child development support offices, which aim to improve access by establishing many offices in different areas (*Ministry of Health Labour and Welfare, 2017b*). In Japan, the child development support centers include public and private centers that aim to ensure the quality of appropriate support regardless of location (*Ministry of Health, Labour and Welfare, 2017a*). Each municipality in Japan has one public child development support office or center, and there are often several private child development support offices scattered throughout the city (*Ministry of Health, Labour and Welfare, 2021a*). When families living in each municipality seek consultation regarding their children's development, most seek help from the public child development support centers (*Ministry of Health, Labour and Welfare, 2021a*; *Ministry of Health, Labour and Welfare,*

*2021b*). The public child development support centers provide a variety of support to make it easier for families to raise their children who have experienced issues since birth with or without a medical diagnosis, such as group medical care within the facility according to the child's characteristics, referrals to specialists and related medical institutions and healthcare facilities, such as the child development support offices (*Ministry of Health, Labour and Welfare, 2021b*).

Infant health examination statuses can vary widely by geographic region, depending on each country's healthcare system, economic situation, and cultural background (*Mianda, Todowede & Schneider, 2023*). For example, in countries with extensive public health insurance, such as Japan and Germany, health checkups are provided almost free of charge, while in countries such as the U.S., where private insurance is the main source of coverage, the cost burden affects the rate of health checkups. In addition, there are large differences in the frequency of medical checkups and age groups covered in different countries (*Hudak & Committee on Child Health Financing 2022*). In Japan, the Maternal and Child Health Law mandates infant examinations at 18 and 36 months of age, and most local governments also provide examinations at 3–4 months and 9–10 months (*Ministry of Health, Labour and Welfare, 2018*). Infant health check-ups, performed by doctors and public health nurses, include screening children for obvious developmental delays and professional consultative opportunities for families to discuss their child's development. If an illness or developmental delays are detected, the children and their families are recommended to visit a hospital or one of the public child development support centers for care and developmental progress reviews. In Japan, when there are problems at birth or obvious illnesses after birth, children are closely followed up by doctors and physical therapists.

Infants and toddlers simultaneously develop gross motor skills, upper limb dexterity, dexterity, coordination, cognition, language, feeding, visual function, and play at various stages (*Fuschlberger et al., 2023*). The content of family consultations varies depending on the living environment of the child and family. Applying the family-centered approach, which is considered best practice, the professionals involved need to provide accurate advice based on the content of consultation and information from the family (*King & Chiarello, 2014*; *McCarthy & Guerin, 2022*). At the public child development support centers, public health nurses and psychologists often provide initial support for children and their families, while physicians, physical therapists, occupational therapists, and speech therapists are involved as necessary. Currently, the work schedule of each professional differs from support center to support center, and professionals are not able to work with facility users as frequently as they would like (*Ministry of Health, Labour and Welfare, 2021a*; *Ministry of Health, Labour and Welfare, 2021b*).

In Japan, the number of surveys on child development support and research on developmental milestones have been gradually increasing (*Uchikawa, Yamoto & Saito, 2023*; *Kato et al., 2023*). However, information from the public child development support centers, which serve as places for initial developmental consultation, is extremely scarce. Nationwide reports describing the main complaints and characteristics of each region, including the consultation content discussed with parents, are currently unavailable. Our research hypothesis was that if the characteristics of the main complaints of parents who

use public child development support centers could be clarified, it would be possible to provide more specialized advice on child development support.

Accordingly, this study aimed to clarify children's gender and age, presence of referral sources, and characteristics of the main complaints from initial consultations between parents of children with developmental delays and professionals working in one of the public child development support centers in a medium-sized city in Tokyo.

## MATERIALS & METHODS

### Participants

In the 10-year period lasting, from April 2013 to March 2023, 1,245 initial consultation interviews were conducted at a public child development support center (operated by a local government) in Tokyo. Before conducting the initial consultation interviews, all participants were asked to complete a preliminary questionnaire that included specific consultation details (*e.g.*, main complaints). In total, 1,241 cases that were not omitted from the preliminary questionnaire were included in the analysis.

The use of smartphones has increased significantly since the 2010s, making the collection of child-rearing information *via* the Internet even more accessible and convenient, while smartphones have become an important source of child-rearing information (*Ministry of Internal Affairs and Communications, 2018*). Therefore, taking into account the situation closer to the family environment in recent years, the survey period in this study was set for a 10-year period.

The study center was located in a city with a population of approximately 130,000 over the past 10 years, and the birth rate ranged from 920 to 1,000 per year.

### Study design

This was a 10-year all-case study of a city in Tokyo. This research study followed an opt-out protocol for consent, with the approval of the Ethics Committee of Fukushima Medical University (authorization number: 2023-025). Public health nurses and psychologists provided an informed consent explanation during the initial consultation to ensure the privacy of the participants and obtained comprehensive consent from all participants to use the information collected during their use of the facility for the study. Detailed information about this study was made available to the target population through posters in the facility.

### Measures

#### Questionnaire

The preliminary questionnaire was a proxy report developed to describe the consultations that the family needed for their child to access the child development support center. This questionnaire covered five items: gender, age (in months), availability of referral sources for the use of the support center, medical diagnosis, and the primary complaints that they wanted to discuss at the initial consultation. All subjects completed this preliminary questionnaire prior to using the facility. The participants were asked to describe their main complaints (specific consultation details) in the greatest detail possible. After the questionnaires were collected, all statements were entered into an Excel file annually by

facility staff and stored. The sampling criteria for this study were those who responded to all five aforementioned items. In this study, the personal information of the data for the study period was removed and the questionnaire data from 1,241 questionnaires that met the criteria were used for analysis.

The main complaint at the time of the initial consultation was analyzed using a qualitative descriptive analysis method of the free descriptive content of all subjects. Descriptions of the main complaints of parents are often described in a mixture of content, with multiple connotations. In doing so, the meaning was supplemented with a minimum of words according to the context and extracted as codes. The extracted codes were classified based on their semantic similarities. As coding criteria for qualitative data in this study, categorization was conducted from the perspective of analyzing matters found as commonalities in the content of consultations on children's growth and development and child rearing.

From the free-form descriptions of the main complaints, 137 codes were extracted and grouped into 13 categories. In accordance with the aforementioned coding criteria for the qualitative data, the words and phrases listed in the specific examples in Table 1 were classified into categories if they were included. To ensure the validity of the analysis, two researchers (YK and ET) with experience in conducting thematic analyses analyzed the data and created a tentative category table, which was shared among all co-authors and revised several times to ensure semantic consistency. One researcher (YK) finalized the category table (Table 1). The categories identified from the analysis were language development, childcare and preschool counseling, hyperactivity/inattention, general developmental issues, conduct problems, consultation regarding rehabilitation services, communication problems, emotional problems, motor development, feeding problems, mental development, sensory bias, and visual function.

## Statistical analysis

The subjects were divided into two groups: those with ($n = 122$) and those without ($n = 1,119$) a medical diagnosis. Comparisons between groups were performed using the $t$, chi-square, and Fisher's exact probability tests.

To examine whether factors related to the main complaints differed between parents of children with developmental delays, with and without a medical diagnosis, logistic regression analysis with forced entry was performed. Gender, age (in months), and the category of each chief complaint were considered independent variables, while with/without a diagnosis was considered the dependent variable. For the analysis, the group with a diagnosis was identified numerically as "1", while the group without a diagnosis as "0". Boys were identified numerically as "0", and girls as "1". Multicollinearity was confirmed using the variance inflation factor, and model fit was examined using the coefficient of determination ($R^2$). All analyses were performed using SPSS statistical package for Windows version 27.0. (IBM Corp., Armonk, NY, USA). Statistical significance was set at $p < 0.05$.
**Table 1 Main complaint categories with the specific examples from the parental interviews.**

| Category | Concrete example |
|---|---|
| Language development | No pointing, slow speech, slurred speech, slurred speech, delayed comprehension, stuttering, the triadic relationship[a] cannot be confirmed. |
| Childcare and preschool counseling | Worried about going to school, not being able to act in a group, not fitting in with group life, not liking group activities, not being good at group activities, worried about future group life, do not want to go to kindergarten, not being able to play together easily in a group, often not following directions. |
| Hyperactivity/inattention | Restlessness, difficulty sitting and waiting, inability to see surroundings, inability to be in the moment, unable to maintain concentration, running around, intense movements, difficulties with verbal directions, unable to follow directions when agitated, unable to sit still and eat, impulsivity. |
| General developmental issues | Delayed development compared to other children in group activities and playing, slow growth, not toilet-trained, bed-wetting every night, not wanting to go to the toilet, not able to change clothes or eat by themselves, not mastering daily routines, difficulties using chopsticks and scissors, reluctant to draw or do crafts. |
| Conduct problems | Tantrums, easily angered, obsessive, takes time to switch scenes, has trouble switching scenes, does not listen, stubborn, throws things, hits, assertive, hands go before words, unusually nervous, lays down when he doesn't like something, explodes when he can't control his emotions. |
| Consultation regarding rehabilitation services | Consultation relating to a disease/diagnosis, rehabilitation, childcare anxiety, discipline, child rearing, childcare consultation, night crying, training to promote development, connection with local institutions, request for developmental examinations, concerned about developmental deviations, request information relating to future consultations. |
| Communication problems | Communication with others through the crane phenomenon, communication at one's own pace, inability to understand social ambience, not understanding others' feelings, lacking in empathy, inability to respond, lack of interest in surroundings, ignoring, not making eye contact, poor social skills, difficulty catching up in conversation, wave one's hand with the palm facing own when to say goodbye, not turning around when called, not responding to calls, not responding to peers, not playing with others of the same age, not able to play with peers, not able to interact with friends, interpersonal relationship difficulties, difficulties with understanding rules, teasing, not being aware of attention, repeating the same thing even after attention, lack of imitation, lack of imagination, difficulty in reading scenes. |
| Emotional problems | Shyness, anxiety with the unfamiliar, odd voice, high levels of nervousness, anxious and crying, self-injurious behaviors, scene keeping, tics, difficulties with a change of environment, panic at irregularity, inability to control feelings, panic when scolded, separation anxiety, mood swings, mental stability, emotional control, a frequent change of mind, panic at negative words, inability to follow rules that are inconvenient, often confused, have a lot of likes and dislikes. |
| Motor development | Inability to sit up, inability to sit for long durations, inability to walk, slow running, falling easily, clumsy hands, weak fingers |
| Feeding problems | Picky eater, allotriophagy, lack of desire to eat, difficulties using eating utensils, strong attachment to specific food, eating things that have fallen to the ground. |
| Mental development | Mental developmental retardation, intellectual developmental retardation, fogginess, learning delays |
| Sensory bias | Difficulty receiving instructions by listening, fear of sounds, finger sucking/nail biting, putting things in the mouth, poor selection of sensory stimuli, panic at certain sounds, sensitivity to smells, sensory hypersensitivity, reactive hypersensitivity to what they see, unwillingness to wear unfamiliar clothing. |

| Category | Concrete example |
|---|---|
| Visual function | Color weakness, difficulty in reading text. |

**Notes.**

The free description contents of the main complaints at the time of the initial consultation, obtained by means of the preliminary questionnaire, were analyzed using the qualitative descriptive analysis method. If the content of the parent's main complaint included connotations related to the concrete example in the table, it was classified in the corresponding category.

[a]Triadic relationship is a concept that describes the relationship between three things: oneself (the child him/herself), others (the parents), and the object (object: toy, *etc.*) with which one shares attention.

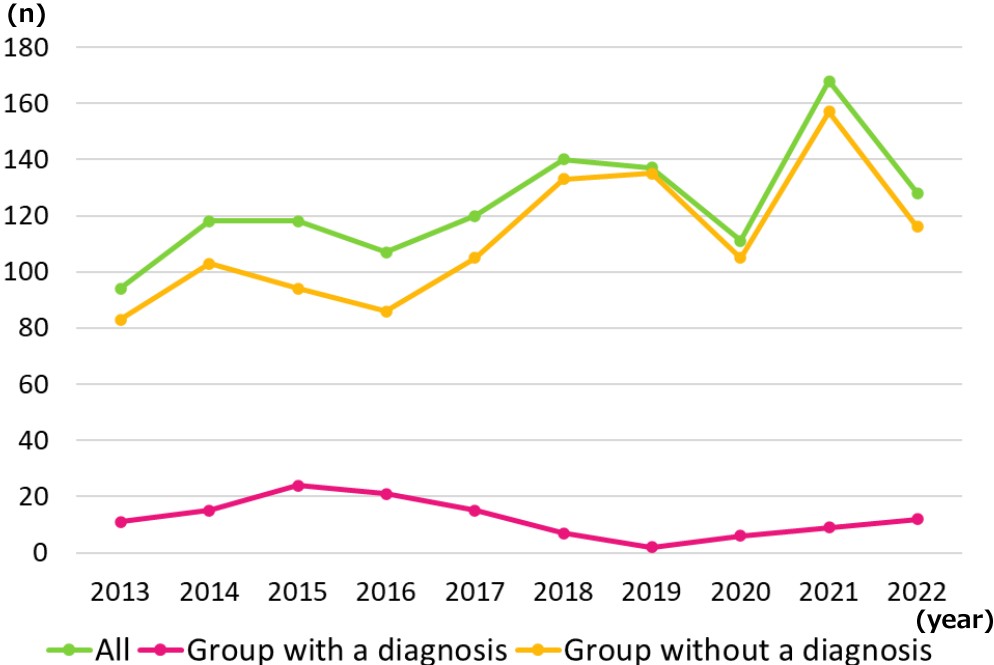

**Figure 1** **Breakdown of the two groups by year.**

# RESULTS

A breakdown of the two groups by year is presented in Fig. 1, while the participant characteristics are listed in Table 2. The proportion of girls was 10% higher in the group with a diagnosis than that in the group without. The mean ages (standard deviation) of the groups with and without a diagnosis were 38.6 (16.7) and 41.1 (17.9) months, respectively ($p = 0.144$). There were no significant differences in the percentages between the two groups in any of the categories. For each category, the percentages of the total number of cases were as follows: Language development: 43.8%, Childcare and preschool counseling: 15.0%, Hyperactivity/inattention: 14.2%, General developmental issues: 14.1%, Conduct problems: 13.9%, Consultation regarding rehabilitation services:13.0%, Communication problems: 12.9%, Emotional problems: 10.5%, Motor development: 8.6%, Feeding problems: 3.2%, Mental developmental: 2.3%, Sensory bias: 1.3%, Visual function: 0.2%. The results of the logistic regression analysis are presented in Table 3. The omnibus test of the model coefficients was significant ($p = 0.044$), while the Hosmer–Lemeshow test

**Table 2  Subject attributes and main complaint categories.**

| | All (n = 1,241) | Group with a diagnosis (n = 122) | Group without a diagnosis (n = 1,119) | p value |
|---|---|---|---|---|
| Gender (boy, girl), n (%) | 905, 336 (72.9, 27.1) | 78, 44 (63.9, 36.1) | 827, 292 (73.9, 26.1) | 0.019* |
| Age in months, month (range) | 40.9 (17.8) (2–87) | 38.6 (16.7) (6–87) | 41.1 (17.9) (2–70) | 0.144 |
| presence of referral sources, n (%) | 944 (76.1) | 94 (77.0) | 850 (76.0) | 0.789 |
| Category of the main complaint, n (%) | | | | |
| Language development | 545 (43.9) | 55 (45.1) | 490 (43.8) | 0.785 |
| Childcare and preschool counseling | 191 (15.4) | 23 (18.9) | 168 (15.0) | 0.264 |
| Hyperactivity/inattention | 172 (13.9) | 13 (10.7) | 159 (14.2) | 0.281 |
| General developmental issues | 169 (13.6) | 11 (9.0) | 158 (14.1) | 0.119 |
| Conduct problems | 166 (13.4) | 11 (9.0) | 155 (13.9) | 0.136 |
| Consultation regarding rehabilitation services | 158 (12.7) | 13 (10.7) | 145 (13.0) | 0.464 |
| Communication problems | 164 (13.2) | 20 (16.4) | 144 (12.9) | 0.275 |
| Emotional problems | 125 (10.1) | 7 (5.7) | 118 (10.5) | 0.094 |
| Motor development | 106 (8.5) | 10 (8.2) | 96 (8.6) | 0.886 |
| Feeding problems | 38 (3.1) | 2 (1.6) | 36 (3.2) | 0.337 |
| Mental development | 31 (2.5) | 5 (4.1) | 26 (2.3) | 0.233 |
| Sensory bias | 15 (1.2) | 1 (0.8) | 14 (1.3) | 0.679 |
| Visual function | 2 (0.2) | 0 (0.0) | 2 (0.2) | 0.640 |

**Notes.**
Average (standard deviation); * vs. group without a medical diagnosis.
*$p < .05$.

indicated a good model fit ($p = 0.730$). The variance inflation factor of the obtained variables was less than 5, and no multicollinearity was present. Regression analysis revealed that the factors associated with the presence or absence of a disease name were gender, age in months, and general developmental issues. The odds ratios (95% confidence intervals) were 1.573 (1.056–2.343) for gender, 0.988 (0.976–1.000) for age (in months), and 0.421 (0.200–0.886) for the general developmental issues. The discriminant accuracy for the presence or absence of a disease name was 90.2%.

## DISCUSSION

As presented in Fig. 1 and Table 2, approximately 90% of families received child development support without a clear diagnosis at the time of the initial consultation. Caregivers' positive emotionality toward their children and avoidance of harsh physical punishment are important for maintaining a positive relationship between caregivers and children, which in turn facilitates children early social and emotional development (*Walker et al., 2011*). Providing early support to anxious parents is important for maintaining

**Table 3 Logistic regression analyses for the presence or absence of a medical diagnosis.**

|  | Coefficient | Wald | Odds ratio | *p* value |
|---|---|---|---|---|
| Constant | −1.396 | 13.873 |  | *p* < 0.001 |
| Gender | 0.453 | 4.964 | 1.573 (1.056–2.343) | 0.026 |
| Age in months | −0.012 | 3.897 | 0.988 (0.976–1.000) | 0.048 |
| General developmental issues | −0.865 | 5.193 | 0.421 (0.200–0.886) | 0.023 |

Notes.

For the analysis, the group with a diagnosis was identified numerically as "1", while the group without a diagnosis as "0". Boys were identified numerically as "0", and girls as "1".

The discriminant accuracy rate for the presence or absence of a diagnosis was 90.2%.

good parent–child relationships. This finding indicates that for families with concerns or difficulties related to child development, support is needed not only for medical care but also for welfare and educational purposes. The most common chief complaint at initial consultation was poor "language development" (43.9%), followed by "childcare and preschool counseling" (15.4%), "hyperactivity/inattention" (13.9%), and "general developmental issues" (13.6%). Children and families are usually not troubled by a single issue but by multiple difficulties (*Ngo et al., 2012*; *Woodman, Mawdsley & Hauser-Cram, 2015*). With limited administrative services, manpower, and time, it is impractical for each professional to individually resolve all issues raised during the initial consultation. This therefore study provides information on the professional knowledge of public health nurses and psychologists, who are often the initial responders to children and families in public child development support centers. Doctors, physical therapists, occupational therapists, and speech-language pathologists involved in child development support need to acquire relevant current knowledge, in addition to using their expertise, to provide appropriate support to children and families under their care.

Specifically, physical therapists and occupational therapists excel in "general developmental issues" and "motor development", and also specialize in working with children who exhibit "hyperactivity/inattention". However, "motor development" was less common, at 8.5% of all consultations. Rehabilitation professionals spend more time with children and their families than physicians, as they engage in educational settings and are involved in implementing long-term environmental adjustments for daily living (*Carey & Long, 2012*). Therefore, professionals involved in child development support, such as public health nurses, psychologists, and rehabilitation professionals, are expected to have a broad knowledge of various developmental issues and comprehensive knowledge of local childcare and schooling systems, in addition to expertise in their specialty, to work effectively with children and families seeking child development support.

Gender, age (months), and general developmental issues were associated with and without a diagnosis. An odds ratio of 1 means that the relationship between the groups with and without a diagnosis is the same, whereas an odds ratio greater than 1 means that the relationship is higher in the group with a diagnosis. Conversely, an odds ratio of less than 1 indicates that the relationship is stronger in the group without a diagnosis. The results of the chi-squared test showed that there was a gender difference between the groups with *vs.* without a diagnosis, and the odds ratio was 1.573, indicating that the

proportion of girls was higher in the group with a diagnosis than in the group without a diagnosis. Children with developmental disorders, such as ASD and autism are more prevalent among boys by about 1:3–4 than girls (*Loomes, Hull & Mandy, 2017*; *Zeidan et al., 2022*). If a congenital condition is detected in utero or at birth, the diagnosis is often confirmed early. However, if a child has a developmental disability, symptoms become more apparent as the child grows (*Zeidan et al., 2022*); therefore, diagnostic decisions are made more cautiously. In many cases, the diagnosis is made around preschool age. In this study, the children's mean age was 40.9 months overall. Many of the children did not have a confirmed diagnosis at the initial consultation. These reasons may have led to a larger proportion of boys in the group without a diagnosis and a larger proportion of girls in the group with a diagnosis.

Since the odds ratio for general development was 0.421, it can be said that consultations related to "general development" were more frequent among children without a diagnosis, and the relationship was higher than for other items. As described in Table 1, free descriptions relating to ADLs included toileting, dressing, and eating, as well as delays in group activities, playing, and the use of tools. These were classified into the general development category. It should be noted that the group without a diagnosis may have more difficulties in daily life. It was suggested that professionals need to be aware of the need to provide information to children without a diagnosis, taking into consideration the common chief complaints in the "general development category".

Child development support in the United States is supported by a number of laws and programs based on the Individuals with Disabilities Education Act and is unique in its emphasis on family support (*Lipkin & Okamoto, 2015*). Specifically, Early Intervention Services, Individualized Education Program, and related services for children with disabilities are provided. Early Intervention Services provides developmental assessment and, as needed, physical, occupational, and speech therapy for children under age 3. The Individualized Education Program is an individualized plan for children over 3 years of age who need special support, and parents, teachers, and specialists work together to develop the plan and address the needs. In addition, "early support" is emphasized in German child development support, and comprehensive programs for children with developmental delays and disabilities from age 0 to school age are practiced (*Benz & Sidor, 2013*). In all countries, including Japan, multidisciplinary involvement is practiced and support in natural living environments, such as home environments and childcare facilities, is emphasized. However, child development support in all countries faces common challenges, such as large regional disparities in support systems, lack of funding, and shortage of professionals, indicating the need to reconsider the nature of child development support in line with social needs (*Ministry of Health, Labour and Welfare, 2022*). To further develop public child development support systems, policymakers or practitioners in each country need to focus on increasing the number of professionals and improving professional education systems to meet the diverse needs of parents.

## LIMITATIONS

The primary strength of this study is that we surveyed all cases monitored over a 10-year period. For all professionals involved in child development support, the 13 categories of the main complaints obtained in this study should be considered as concerns that parents have before consulting a specialist. However, this study has some limitations. To avoid subject selection bias, this 10-year case study was conducted a city in Tokyo. However, parents who are being followed up in the hospital immediately after birth due to birth or postpartum problems sometimes do not use public developmental centers. Therefore, group with a medical diagnosis may be slightly less than the original population. Also, the situation may be unique to the location and different to regional cities within the same country. Thus, the generalization of the results to the situation in Japan as a whole may be limited. Additionally, this study used a questionnaire completed by parents prior to the initial interview. It is possible that the main complaints directly elicited by professionals during the interviews contained more information. Therefore, future studies are needed to investigate parental main complaints by region and to investigate longitudinal changes over time, such as whether the main complaints of parents change as their children grow older.

## CONCLUSIONS

At the time of the initial consultation, 90.2% of the families had received child development support without a clear diagnosis. The most common chief complaint was "language development" (43.9%), followed by "childcare and preschool counseling" (15.4%), "hyperactivity/inattention" (13.9%), and "general developmental issues" (13.6%). In contrast, "motor development" was less commonly raised as a concern (8.5% of all consultations). Professionals involved in child development support are expected to have broad knowledge of various developmental issues and comprehensive knowledge of local childcare and schooling systems. As such, to further develop public child development support systems, policymakers or practitioners in each country need to focus on increasing the number of professionals and improving professional education systems to meet the diverse needs of parents.

## ACKNOWLEDGEMENTS

We would like to thank Editage for professional English language editing. The authors would further like to thank all the participants of this study.

### Funding

This study was financially supported by the JSPS KAKENHI (grant number: 20K02710). The funders had no role in study design, data collection and analysis, decision to publish, or preparation of the manuscript.

## Grant Disclosures

The following grant information was disclosed by the authors:
JSPS KAKENHI: 20K02710.

## Competing Interests

The authors declare there are no competing interests.

## Author Contributions

- Yasuaki Kusumoto conceived and designed the experiments, performed the experiments, analyzed the data, prepared figures and/or tables, authored or reviewed drafts of the article, and approved the final draft.
- Eri Takahashi conceived and designed the experiments, performed the experiments, analyzed the data, authored or reviewed drafts of the article, and approved the final draft.
- Kenji Takaki conceived and designed the experiments, performed the experiments, authored or reviewed drafts of the article, and approved the final draft.
- Tadamitsu Matsuda conceived and designed the experiments, performed the experiments, authored or reviewed drafts of the article, and approved the final draft.
- Osamu Nitta conceived and designed the experiments, performed the experiments, authored or reviewed drafts of the article, and approved the final draft.

## Human Ethics

The following information was supplied relating to ethical approvals (*i.e.*, approving body and any reference numbers):

The Fukushima Medical University granted ethical approval to conduct the study within its facilities (authorization number: 2023-025).

## Data Availability

The data are available at Figshare: Yasuaki, Kusumoto (2024). Main complaints identified by parents of children with developmental delays during the initial consultation: A 10-year all-case study. figshare. Dataset. https://doi.org/10.6084/m9.figshare.28106102.v1.

## Supplemental Information

Supplemental information for this article can be found online at http://dx.doi.org/10.7717/peerj.19044#supplemental-information.

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
