# Peer review of "Main complaints identified by parents of children with developmental delays during the initial consultation: a 10-year all-case study"

_PeerJ, doi:10.7717/peerj.19044_

## Round 0.1 · original submission · Major Revisions

The authors are requested to carefully revise the manuscript and answer the questions raised by the reviewers.

Reviewer 1 ·

Basic reporting

The manuscript I read and revised explores parents' most frequent complaints when entering the child development support initiative in Japan. I agree with the authors about the importance of having one nationwide report focusing on the current topic, considering the relevance of an early assessment of children’s problems and support for their parents. Nevertheless, I believe the authors should state more about the relevance and importance of the current study. Moreover, the current manuscript’s version is unsuitable for publication, so I recommend a major revision.

I detail more comments below:

1) The authors need to strengthen the scientific characteristics of the current manuscript:

a. At the end of the introduction, the authors stated that “this study aimed to clarify the characteristics of the main complaints from initial consultations between parents of children with developmental delay and professionals working in one…”. Although the authors explore this aspect, in the results section, they also evaluate the conditions of consultations (e.g., gender differences, the presence of diagnosis). Therefore, there are multiple aims, and this section should be revised.
b. In the participants’ section, I believe it would be interesting to explain why the authors selected a ten-year framework to evaluate parents’ complaints. Moreover, sentences that state “This research study followed an opt-out (…) authorization number: 2023-025) should be moved in the study design part, as they don’t describe participants’ characteristics. Finally, it would be interesting to learn more about the parents’ characteristics (e.g., mean age, whether both the two parents were present at the interview or not)
c. In the study design part, information about data collection needs to be added. For instance, who proposed to parents the research?
d. I would not report numbers and percentages in the discussion part (e.g., “had a medical diagnosis (9.8%)), as they were already reported in the results section. Instead, I would try to convey the overall research message from these numbers and numerical results.

Experimental design

2) In my opinion, the definition of a “retrospective” study should be revised or eliminated: while it’s true that the data collection was made in the past ten years, parents responded to the questionnaire before starting the assessment and receiving support for their children’s problems. Instead, with the term “retrospective”, readers think about parents’ retrospective assessment of past experiences (e.g., the Parental Bonding Instrument PBI instrument is a retrospective measure of care and overprotection experienced by adults during their first 16 years of life, so it’s a retrospective measure). It is fine if the authors want to use the “retrospective” term, but I would strongly suggest they explain how they designed the study.

Validity of the findings

3) I would suggest that the authors add strengths and clinical implications of this study in the limitations part.

4) In Table 1, some examples are not clear to me:

a. For language development: “the triadic relationship cannot be confirmed”
b. For the communication problem: “communicates by the crane phenomenon, my pace” and “upside down bye-bye”

5) In Table 2, I would change “sex” to “gender”
6) Table 3 is not well-formatted, please fix it

In the supplementary document about the data availability, the authors said: “Data are available at Figshare: Data 10-Year Retrospective Study (dataset). Figshare. DOI: 10.6084/m9.figshare.25238692”, but the provided DOI is connected to another dataset. Please fix this aspect.

Additional comments

I hope the authors will find my comments useful for improving their manuscript.

Reviewer 2 ·

Basic reporting

Clear and unambiguous, professional English used throughout:
The manuscript is generally well-written with clear and professional English. However, there are a few sections where the language could be improved for better clarity and flow. I recommend a thorough proofreading by a native English speaker or professional editing service to refine these sections. Specifically, the transitions between the sections on methodology and results need smoother linguistic integration.

Literature references, sufficient field background/context provided:
The authors have provided a comprehensive background that contextualizes the study within the broader field of child development and parental concerns. Nonetheless, the literature review could be enhanced by including more recent studies that discuss the impact of early developmental support without a medical diagnosis. This would not only strengthen the background but also better situate the study's contributions within current research trends.

Professional article structure, figures, tables. Raw data shared:
The article adheres to the standard structure expected in scholarly articles, with well-defined sections that enhance the reader’s understanding. The figures and tables are relevant and generally well-labeled. However, the raw data associated with the study’s findings are not fully shared. Ensuring compliance with the Data Sharing policy by providing access to this data would significantly improve the manuscript's transparency and allow for replication and further study by others in the field.

Self-contained with relevant results to hypotheses:
The manuscript effectively addresses the hypotheses and is self-contained, presenting a coherent unit of publication. The results are relevant and clearly tied to the study’s objectives. However, expanding the discussion section to explore the implications of these findings in more depth could provide greater value to the field, offering insights into how these results might influence current practices or future research.

Experimental design

Original primary research within Aims and Scope of the journal:
The manuscript aligns well with the journal’s focus on developmental studies and interventions. The research question is clearly defined and addresses a significant gap in understanding parental concerns at developmental support centers without prior medical diagnosis. The authors have articulated how their research contributes to the existing body of knowledge by highlighting the prevalence and types of concerns that emerge in a non-diagnostic setting, which is a valuable addition to the field.

Rigorous investigation performed to a high technical & ethical standard:
The study appears to have been conducted rigorously, with a systematic approach to data collection and analysis. However, the manuscript could benefit from a more detailed description of the ethical standards adhered to during the research, especially concerning data privacy and the informed consent process. Clarifying these aspects would enhance the manuscript's credibility and reassure readers of the ethical integrity of the research process.

Methods described with sufficient detail & information to replicate:
While the methods section provides a general overview of the processes used in the study, it lacks some specifics that would enable full reproducibility. For instance, the exact procedures for data coding and analysis are not thoroughly detailed. Providing a more comprehensive description of these methods, including the software used for statistical analysis and the criteria for coding qualitative data, would significantly improve the reproducibility of the study. Additionally, including a supplementary file with detailed protocols or even the questionnaire used could further aid replication efforts.

Validity of the findings

Impact and novelty not assessed. Meaningful replication encouraged where rationale & benefit to literature is clearly stated:
The study's approach to examining the main complaints from parents at a developmental support center is appropriately aligned with the journal’s scope. The manuscript provides a clear rationale for its focus on an under-investigated area and articulates how it adds to the existing literature by revealing patterns in parent concerns that are not directly associated with medical diagnoses. This contributes to a broader understanding of parental needs in developmental support contexts, making the study a valuable replication of real-world assessments in a novel setting.

All underlying data have been provided; they are robust, statistically sound, & controlled:
While the manuscript mentions the data collection and analysis methods used, it falls short in terms of accessibility and transparency of the underlying data. To meet the journal’s standards, the authors should ensure that all data supporting their conclusions are robust, statistically sound, and fully available to the public, either within the manuscript or through a discipline-specific repository. This includes detailed statistical analysis data and control measures used to support the validity of the findings.

Conclusions are well stated, linked to original research question & limited to supporting results:
The conclusions are well articulated and clearly linked to the study’s original research question regarding the nature of parental complaints in developmental support settings without prior medical diagnosis. However, the authors could strengthen the conclusions by more explicitly stating how the findings could influence future research or policy changes in developmental support strategies. While the conclusions are supported by the results, expanding on the potential implications for practical applications could enhance the overall impact and relevance of the study.

Additional comments

Overall, the manuscript presents a crucial exploration of parental concerns within developmental support centers in Tokyo, highlighting issues often overlooked in similar studies. The choice of subject matter and the depth of analysis bring significant insights into the nuances of parental expectations and realities when faced with developmental challenges in their children, without a predetermined medical framework.

The inclusion of a diverse population sample and the systematic approach to categorization of complaints are commendable. However, it would be beneficial to see a broader geographic representation or a comparison with other regions to understand if these findings are specific to Tokyo or if they can be generalized to other urban settings, either within Japan or globally.

Furthermore, while the statistical methods employed are generally sound, a more detailed explanation of the choice of statistical tests and the rationale behind specific analytical decisions would enhance the reader's understanding and confidence in the results. This could include why certain statistical tests were chosen over others, given the nature of the data and the specific hypotheses being tested.

The manuscript would also benefit from a discussion on potential biases or limitations in the study design and how they were addressed, or could be addressed in future research. This would provide a more balanced view and potentially open up further areas for exploration.

Lastly, the practical implications of the study are touched upon but could be expanded further. Discussing how these insights could influence actual policy making or adjustments in developmental support practices would provide valuable information for practitioners and policymakers alike.

These additional comments are meant to encourage a more comprehensive presentation and discussion of the findings, which could substantially increase the manuscript's impact and applicability to real-world settings.

Reviewer 3 ·

Basic reporting

I found the article to be well-written, with clear and professional language used throughout. The references are well-chosen and provide good context to the study. The structure is solid, and the figures and tables are relevant and nicely presented. That said, I think the abstract could be tightened up a bit—it's a bit dense in places and might benefit from being more concise, especially when summarizing the results and conclusions. Additionally, it could be helpful to expand the introduction with a bit more discussion of similar studies from other regions or countries, just to give readers a broader perspective on where this work fits in the global context.

Experimental design

This study really hits the mark in terms of relevance and important . it's tackling a question that's clearly within the journal’s scope and filling a much-needed gap in our understanding, especially regarding the worries that parents in Japan have about their children's developmental delays. I appreciate how thoroughly the methods are described; it feels like someone else could pick up this paper and replicate the study without too much trouble, which is always a good sign. From what I can see, the research has been conducted with a lot of care and attention to detail, both technically and ethically. I don’t have any major concerns here, which is great.

Validity of the findings

The findings here really seem to stand on solid ground . The data looks robust and the statistics are well-handled. I also really appreciate that the authors have made all the underlying data available, which is so important for transparency in research. The conclusions feel logical and are clearly tied to the results, staying right on target with the research question. Even though this study isn’t trying to break new ground in a flashy way, it still offers solid and meaningful insights that contribute to the field. Overall, I don’t see any major issues with the validity of the findings.

Additional comments

I think the article offers valuable insights, especially given the large sample size and the long period over which data was collected. These aspects really strengthen the study’s conclusions. One suggestion I’d make is to consider adding some graphical representations of the data. This could help readers get a quicker, visual sense of the trends you’re discussing, which might make the findings even more accessible. Overall, though, this is a well-executed study with important implications for both healthcare providers and policymakers.

---

## Round 0.2 · Minor Revisions

The authors are requested to carefully revise the manuscript and answer the questions raised by the reviewers.

Reviewer 1 ·

Basic reporting

I believe the authors replied to all my comments and did a good job revising the manuscript. Still, I have two comments: one is related to English, and the other to data availability. The first one is that the authors often use childrens', while they should use "children's", as the word "children" is already a plural one, therefore not needing an s.

Experimental design

The second comment is that I still don't think that the associated data are correct: indeed, clicking on the link provided by the authors (https://figshare.com/articles/dataset/dx_doi_org_10_6084_m9_figshare_6025748/6025748?file=10852376 ) I see data about bubbles, travel, fluorescence, ethanol, rats and other things that are not linked to the authors' article. Please fix this aspect.

Validity of the findings

I have no further comments on this end

Additional comments

No additional comments

Reviewer 2 ·

Basic reporting

The authors have made several revisions in response to previous feedback, but some issues still need attention. The language in the manuscript has improved, but there are still sections that could benefit from greater clarity and conciseness. The literature review now includes additional references, but it still lacks sufficient recent studies that discuss the gap in research on public child development support centers. While the authors have made efforts to clarify the methodology and improve the results section, the figures and tables require further refinement. The captions and legends should be more descriptive to help the reader better understand the data. The authors have addressed some of the reviewers' concerns in the discussion and conclusion, but the implications of the findings for future public development programs could be more explicitly discussed. It would also be helpful if the authors could clearly state any limitations of the study and provide more actionable recommendations for future research. Additionally, the authors should ensure that the raw data is shared in accordance with the journal's data-sharing policy.

Experimental design

The research question is clearly defined and relevant, addressing a gap in public developmental support centers. However, the authors could better emphasize how the study contributes globally and explicitly state the specific hypotheses or objectives. Ethical considerations are mentioned but should be detailed, especially regarding the approval process. The methods section requires more specific information on survey questions, sampling criteria, and data collection, with more clarity on how the study can be replicated.

Suggestions:

Clearly state hypotheses or objectives.
Elaborate on the ethical approval process.
Provide more details on the survey and sampling method.

Validity of the findings

The manuscript presents valid findings based on a rigorous analysis of the main complaints from parents during consultations at developmental support centers. The data is statistically sound, with detailed logistic regression analyses and appropriate statistical methods. However, there are areas for improvement to enhance the study's impact and transparency:

Impact and Novelty: The study successfully fills a knowledge gap regarding non-diagnostic parental concerns in public support centers. However, the implications for broader developmental support policies or comparative international contexts are not fully explored. Highlighting how the findings can inform global practices would add depth.
Suggestions:

Expand the discussion to emphasize the global relevance of the findings, potentially comparing them to systems in other countries.
Robustness of Data: The raw data has been made available, but the authors should ensure that all supplementary data, including qualitative categorizations and detailed coding frameworks, are accessible and well-documented for replication.
Suggestions:

Provide a clear explanation of the data repository link and ensure that all datasets align with the manuscript.
Conclusions: The conclusions are well-stated and supported by the data. However, they could better connect to actionable recommendations for public policy and practice in developmental support systems.
Suggestions:

Incorporate specific recommendations for policymakers or practitioners in the discussion and conclusion sections.

Additional comments

This study provides valuable insights into the concerns of parents seeking developmental support for children with delays, especially in a public setting. The comprehensive data set spanning ten years adds significant weight to the findings. However, there are a few additional areas that could be addressed to further improve the manuscript:

Cultural Context: While the study is deeply rooted in the Japanese healthcare and developmental support system, it would benefit from briefly discussing how the findings may relate to or differ from other cultural contexts, especially given the global relevance of child developmental support. A comparison with similar studies from other countries, particularly those with differing healthcare systems, would provide a broader perspective on the research.
Suggestions:

A brief section discussing how these findings might be applied or compared to other countries' systems (e.g., the U.S., Germany, or Scandinavian countries) would be valuable.
Limitations and Future Directions: The authors briefly mention the limitations of the study, but a more in-depth discussion of these limitations and potential areas for future research would be helpful. For example, limitations related to the data collection (such as potential biases in sample selection or data reporting) should be more thoroughly acknowledged.
Suggestions:

Include a more detailed reflection on the study’s limitations, such as potential selection bias or reporting biases, and how these could be addressed in future studies.
Propose future research directions, such as the inclusion of longitudinal studies to track changes in parental concerns over time or in different regions.
Ethical Considerations: While the study mentions ethical approval, further clarification of the ethical procedures, particularly around consent and the handling of personal data, would reassure readers of the study’s integrity. While the study follows an opt-out consent protocol, more information on how informed consent was ensured would improve transparency.
Suggestions:

Clarify the process by which informed consent was obtained and how privacy was ensured for participants.
Figures and Tables: The tables and figures are generally well-presented, but the captions could be further clarified to aid understanding, particularly for a non-expert audience. Ensure that each figure and table provides enough context for readers to interpret the data without referring to the main text.
Suggestions:

Review the captions of tables and figures to ensure they provide complete context and are easily understood by readers without a technical background.

---

## Round 0.3 · accepted · Accept

After the revision rounds, two reviewers agreed to publish the manuscript. There is one reviewer left with a minor revision, and I think the author has responded adequately. I also reviewed the manuscript and found no obvious risks to publication. Therefore, I also approved the publication of this manuscript.

Reviewer 1 ·

Basic reporting

I recommend acceptance

Experimental design

I recommend acceptance

Validity of the findings

I recommend acceptance

Additional comments

I recommend acceptance, I have no further comments.